# Fabrication of Silver Iodide (AgI) Patterns via Photolithography and Its Application to In-Situ Observation of Condensation Frosting

**DOI:** 10.3390/nano13233035

**Published:** 2023-11-28

**Authors:** Takao Okabe, Jinchen Tang, Katsuhiko Nishimura, Naoki Shikazono

**Affiliations:** 1Institute of Industrial Science, The University of Tokyo, Komaba, Meguro-ku, Tokyo 153-8505, Japan; kappon@iis.u-tokyo.ac.jp (K.N.); shika@iis.u-tokyo.ac.jp (N.S.); 2Institute of Engineering Thermophysics, Chinese Academy of Sciences, Beijing 100190, China

**Keywords:** photolithography, nanoparticle, patterning, frosting, in-situ

## Abstract

This study introduces an innovative photolithography-based method for patterning ionic and inorganic particle materials such as silver iodide (AgI). Conventional methods lack precision when patterning powdered materials, which limits their applicability. The proposed method stacks layers of a particle material (AgI) and negative-tone photoresist for simultaneous ultraviolet exposure and development, resulting in well-defined AgI patterns. The sintering process successfully removed binders from the material layer and photoresist, yielding standalone AgI patterns on the Si substrate with good adhesion. The pitch remained consistent with the design values of the photomask when the pattern size was changed. In-situ observation of condensation frosting on the patterns was conducted, which confirmed the practicality of the developed patterning process. This versatile method is applicable to large areas with a high throughput and presents new opportunities for modifying functional surfaces.

## 1. Introduction

There has been increasing demand to develop technologies for fabricating microscale patterns on the surfaces of materials in surface engineering [1,2,3], optical [4,5], microfluidic [6,7,8], and thermal engineering [9,10,11,12,13,14,15], etc. For example, micropatterning has been proposed to utilize the lotus leaf effect, which prevents the adhesion of water droplets and dirt by increasing the contact angle [1,2,3]. Micropatterning has also been applied to optical lenses to achieve antireflective effects [4,5]. The number of studies on patterning has increased, and these include methods such as culturing cells on polymer dot patterns [6] and creating microchannels for analyzing the movement of microscale fluids [7,8]. In thermal engineering, the evaporation behavior of water on micropatterns has been studied to realize efficient heat exchange [9,10,11]. Furthermore, patterned surfaces are expected to alter frost formation [12,13,14,15], which is an important phenomenon in heat exchange and refrigeration.

Most patterning processes such as photolithography [16,17] and nanoimprint lithography [18,19,20,21] were developed for patterning polymer surfaces, making it challenging to apply them to the fine patterning of inorganic and ionic materials. An example of patterning materials other than polymers is a method of patterning ceramics using thermal imprinting, wherein a mixture of ceramic particles and a thermosetting polymer is heated and transferred to a mold [18,19]. Another example is the ultraviolet (UV) nanoimprint method, which uses mixtures of UV-curable resin and ceramic powders for duplicating micropatterns [20,21]. In both methods, a patterned material surface can be obtained after sintering.

In frost research, there is growing interest in patterning ionic crystal materials such as patterning silver iodide (AgI) or calcite (CaCO_3_) to control frost formation [22,23,24]. These materials are selected because their crystal shapes are similar to those of the H_2_O ice crystals. Frost formation involves complicated processes, and therefore, nano- and microscale frost formation and growth behaviors are of fundamental interest. Patterning frost-inducing substances helps to observe the initial behavior of frost growth on the created patterns on a microscale. Aizenberg et al. fabricated precipitated CaCO_3_ patterns using a soft material (polydimethylpolysiloxane, PDMS) stamp on a substrate [23]. Liu et al. deposited random AgI dots via the chemical reaction of KI and AgNO_3_ on a substrate to demonstrate the relationship between the frost shape and substrate wettability [24,25].

However, pattern deposition using aqueous solutions is limited by the area of the pattern and its resolution. In thermal and UV imprint lithography, the entire substrate is covered by a single material layer (referred to as a residual layer) [20,21], which makes it challenging to obtain the pattern of a material different from the substrate, as shown in Refs. [23,24,25]. Techniques such as sputtering and vapor deposition can create material patterns; however, they alter the crystal structures of the materials. Patterning by etching the surface with corrosive liquids or electrical corrosion is another such approach, but it poses challenges in terms of the potential roughening of the entire surface and the inability to control substrate surface roughness [26]. A high-resolution, large-area patterning method for ionic crystals that is unaffected by the material and roughness of the substrate is required to advance frost research.

To overcome these challenges, we propose a versatile patterning method for ionic material particles (e.g., AgI) that stacks the photoresist and particle layers and simultaneously exposes and develops both layers. This method allows the direct patterning of ionic materials on substrate materials without a residual layer. Therefore, this method can create arbitrary patterns without sputtering or vapor deposition, making it practical and suitable for diverse fields, including frost research.

We focus on the fabrication of AgI patterns for the in-situ observation of condensation frosting. To the best of our knowledge, no previous studies have been conducted on the fabrication of patterned AgI. We explored the AgI pattern fabrication process, which includes the slurry conditions of AgI particles for creating AgI layers and the reproducibility of the pattern dimensions. In addition, we observed local frost formation on the fabricated AgI patterns, highlighting the practicality and utility of the developed patterning process. 

This paper consists of an explanation of the proposed process and its validation through experiments. Section 2 provides a detailed description of the process and materials involved in the proposed method. In addition, an explanation of in-situ observation method of condensation frosting using a sample created via the proposed method is described. Section 3 presents the results of an investigation into the material conditions required for AgI patterning, and verifies whether the processes of exposure and development proceeded as expected. It also includes a verification of the difference between the pattern after sintering and the design value of the photomask, and a discussion on the reasons for this discrepancy. Finally, by demonstrating in-situ observation of frost growth at a micro level using the fabricated AgI pattern, the utility of the AgI pattern for frost research is demonstrated. Section 4 concludes with a discussion on the usefulness and challenges of the proposed method, prospects, and potential applications.

## 2. Experimental

### 2.1. Proposed Process

In the proposed photolithography method, the resist is exposed through a material layer stacked on the photoresist, as shown in Figure 1a. The material layer is composed of material particles and a binder. The binder serves to connect the particles with each other, forming a thin layer. This structure allows for the creation of a cohesive and stable material layer. During the development process, the two stacked layers are developed together, as shown in Figure 1b. As drawn in Figure 1b, AgI particles on unexposed areas are removed by a developer. After development, a pattern consisting of two stacked layers—a material layer and a resist layer—is obtained. Two important points are considered in this study: 

(1)The material layer must have clearance between particles, allowing UV light to pass through. (2)A binder of material particles that are insoluble in the developer solution must be selected to prevent the loss of all material layers during development.

### 2.2. Materials and Methods

The material layer was formed using a slurry of AgI particles. Initially, AgI particles (10.0 g) (7783-96-2, Kanto Chemical Co., Inc., Tokyo, Japan) were subjected to wet-ball milling for 2 months to reduce their diameter to less than the minimum dimension of the photomask. Amounts of 6.7 g and 40.6 g of zirconia balls with diameters of 2 and 5 mm, respectively (Toray Industries, Inc., Tokyo, Japan), and 20 g of ethanol (64-17-5, Kanto Chemical Co., Inc.) were mixed in a Teflon container. After the mixing, the Zr balls were removed, and the ethanol was evaporated via heating in a furnace at 70 °C on a glass Petri dish. Subsequently, the dried AgI was subjected to additional milling using a mortar and pestle. The particle size distribution was measured using laser diffraction measurements (Partica LA-960V2, HORIBA, Ltd., Tokyo, Japan), and particles smaller than 1 µm were confirmed using scanning electron microscopy (SEM).

A binder was prepared, which was a mixture of ethyl cellulose (Kanto Chemical Co., Inc.) and a terpineol solvent (98-55-5, WAKO, Tokyo, Japan), with a mixture ratio of 3 and 97 wt.%, respectively. The ethyl cellulose facilitates connections between nanoparticles, sustaining the particles through resistance development and sintering. The details of the binder and slurry conditions have been reported in references [27,28,29]. The milled AgI nanoparticles were mixed with a binder at weight ratios of 1:1, 1:2, and 1:3. The mixture of the AgI nanoparticle and binder underwent two rounds of mixing at 1600 rpm for 180 s using a rotational mixing machine (KK-250S, Kurabo, Osaka, Japan). The wettability of the slurry on the resist layer could have affected the formation of the material layer; therefore, the wettability performance was evaluated by measuring the contact angles (DMo-501, Kyowa Interface Science Co., Ltd., Saitama, Japan) for the three slurries. The contact angles of the slurries were measured from the image of a droplet on the before-UV exposed SU8 with a volume of 2 µL on the resist layer.

The details of the proposed fabrication process are illustrated in Figure 1c. Approximately 100 µL of resist was applied to a 25 mm square-cut Si wafer (c-1). A negative tone resist, SU8 (SU8-3050, Kayaku Advanced Materials, Westborough, MA, USA), was used for the photoresist, and forms micro- and nanoscale patterns with thick resist layers. The applied SU8 was spin-coated using a spin coater (POLOS Spin150i SPS, Putten, The Netherlands) at a top spin speed of 10,000 rpm for 80 s (c-2). The coated SU8 layer was pre-heated on a hot plate at 95 ℃ for 10 min [30] to eliminate the solvent in SU8 and pre-harden the layer (c-3). Subsequently, the prepared AgI slurry was spin-coated on top of the SU8 layer with a spin speed of 10,000 rpm for 80 s, thereby forming the crucial material layer (c-4). Immediately after slurry coating, the substrate was heated on the 95 °C hot plate for 10 min to remove terpineol from the AgI slurry (c-5). After evaporation, AgI nanoparticles bound to ethyl cellulose formed an AgI particle layer as a material layer on the pre-exposed SU8 layer. In the UV exposure process (c-6), SU8 was exposed to UV light (UP50, Panasonic Corporation, Osaka, Japan) with a main spectral peak at 365 nm passing through a photomask and material layer for 1 min. The photomask was designed to include six areas, which were three dotted and three line areas, as shown in Figure 2. The dot diameter and line width were both set to 50 µm, and three different distances were prepared as pitches of 200 µm, 400 µm, and 800 µm. The black areas in Figure 2 represent areas covered with a Cr mask because a glass-type photomask was used; the white areas are transparent and exposed to UV light. For SU8, which is a negative-tone resist, the white and black areas correspond to the cured and developed areas, respectively. Following the exposure and removal of the photomask, the sample was reheated on the 95 °C hot plate for 10 min for the chemical reaction of SU8 under dark conditions. This heating is necessary for SU8, while nonchemically amplified resists do not require post-exposure heating [30]. After UV exposure, the sample underwent wet development by being dipped in the developer for 5 min (c-7), which results in the formation of a stacked layer pattern consisting of AgI nanoparticles and an SU8 patterned layer (c-8). Propylene glycol 1-monomethyl ether 2-acetate (PGMEA) (108-65-6, Kanto Chemical Co., Inc.) was selected for the wet-etching of SU8 [31,32]. After development, the samples were rinsed with pure water. As depicted in Figure 1d, PGMEA did not dissolve the ethyl cellulose within the development time, thereby maintaining the AgI layer on the exposed SU8. Then, crystallized AgI patterns were created by sintering the sample in a furnace (MMF-1W, AS ONE Corporation, Osaka, Japan) at a maximum sintering temperature of 480 °C for 3 h with a heating rate of 10 °C/min (c-9). The heating program is shown in Figure 1e, and the sintering process was conducted under atmospheric conditions. The maximum sintering temperature was deliberately kept lower than the melting point of AgI (552 °C) [33] and the sintering temperature reported in Ref. [34] because of the nanoscale nature of the particles, which easily sinter at lower temperatures. The sintering temperature needs to be above 350 °C to sufficiently remove SU8. If the temperature is below 350 °C, there is a possibility that carbonized resist may remain, depending on the type of resist. During the sintering process, both the resist and ethyl cellulose were completely removed, leaving behind the AgI crystalline patterns (c-10). After maintaining the maximum temperature of 480 °C for 3 h, the temperature was naturally decreased to room temperature. The sizes of the patterns, both before and after sintering, were measured using an optical microscope (VK-X1000, Keyence, Osaka, Japan) and SEM with EDS (TM3030, Hitachi High-Tech Corporation, Tokyo, Japan) to investigate the changes in the patterning dimensions through the sintering process. An EDS mapping image was measured by the EDS for the post-sintering sample to clearly show the AgI pattern forming on the Si substrate without any decomposition of AgI occurring.

To investigate the strength of the AgI pattern after sintering, a peel-strength test, also known as a tape test, was conducted using adhesive tape. The tape (P-221, Nitto Denko Corporation, Tokyo, Japan), with an adhesion force of 278 g/10 mm, was fully adhered to the AgI pattern, and, by peeling it off, we confirmed the adhesion strength of the sintered AgI to the Si substrate based on the presence or absence of pattern peeling.

In addition, this paper demonstrates in-situ observation of frosting using the fabricated AgI pattern. The fabricated sample was placed on a plate cooled using a Peltier device. A camera (HY-2307, HAYEAR, Shenzhen, China) with a frame rate of 60 fps and full HD resolution along with an objective lens at 100× magnification (M Plan Apo SL100×, Mitutoyo Corporation, Kawasaki, Japan) was positioned at the top of the sample. Illumination was provided by a ring light and the focus was adjusted to the Si wafer surface. In observations using a dot pattern, frost was generated under atmospheric exposure, with the room temperature and humidity during the experiment being 25 °C and 35% RH, respectively. The temperature on the Si wafer of the sample was −10 °C. For observations using a line pattern, a frost generation device and optical microscope (VW-9000SP, Keyence, Osaka, Japan) were introduced into a constant temperature and humidity chamber (PL-3KP, ESPEC Corp., Osaka, Japan). The temperature and humidity in the chamber were 23.9 °C and 36.0% RH, respectively. The temperature on the Si wafer of the sample was −9.0 °C.

## 3. Results and Discussion

### 3.1. Particle Size and Slurry Wettability Optimization

We investigated the particle size required to adjust the slurry because the smallest particle size determines the minimum pattern size. Figure 3a shows the particle size distribution obtained. The particle distribution indicates a reduction in size after milling, with the peak shifting from 10 µm (black) to 8 µm (red), and the second peak in the original AgI pattern (black) disappearing at ~200 µm. Thus, particles exceeding the minimum dimensions of the mask pattern (dotted line in Figure 3a) were removed, which ensured that no particles had diameters beyond the designed pattern size. 

The SEM images shown in Figure 3b,c indicate a reduction in particle size after milling. The SEM image in Figure 3c indicates a minimum particle size of 300 nm, which is significantly smaller than the minimum photomask pattern size of 50 µm. This suggests that these particles are suitable for lithography. A smaller particle size is advantageous for slurry dispersion. In this experiment, slurries with ratios of 1:1, 1:2, and 1:3 were prepared, and they showed no visible separation for approximately 7 days at each mixing ratio.

The wettability of the slurry on the resist layer is crucial for forming the material layer. A low-viscosity slurry, ideal for spin coating, can have low wettability on smooth surfaces such as the spin-coated resist layers [35]. Furthermore, there is a lack of prior research on slurry wettability under the proposed conditions. The wettability performance was evaluated from contact angles at three different concentration ratios. Figure 4 shows the contact angles of the AgI slurry droplets with AgI-to-binder ratios of 1:1, 1:2, and 1:3, and the binder on the SU8 layer before UV exposure. As shown in Figure 4, all slurries displayed small contact angles, with the 1:1 slurry having the smallest contact angle of 27.9°. Slurries with higher binder ratios exhibited larger contact angles. Although all slurries were able to create an AgI layer on SU8 by spin coating, the 1:1 ratio slurry emerged as the most suitable for this process because of its excellent adhesion and uniform thickness. Therefore, a 1:1 ratio was used for sample fabrication in this study.

### 3.2. Fabrication

A photograph of the developed sample is shown in Figure 5. As seen in Figure 5a, the light-yellow dots and line patterns, both displaying the color of AgI, are clearly visible on the Si substrate after the development process. No noticeable pattern losses or unexposed spots are observed in any of the patterned areas. Figure 5b shows the SEM image of the pattern before sintering, with both dot and line patterns visible. The pattern on the surface appears as a mottled black and white pattern, which is clearly visible on the line pattern. The black areas are either parts where SU8 is exposed or areas with a low density of AgI. The white areas are where AgI particles are placed on top of the SU8. Although the distribution of AgI was not uniform, it can be seen that AgI was placed on the pattern after development process.

The ability of the AgI layer to remain on the pattern without dissolving into the developer is attributed to the insolubility of ethyl cellulose in PGMEA, as depicted in Figure 1d. As seen in Figure 1d, ethyl cellulose remained in PGMEA within the 5-min development time. As also described in Section 2.1, the binder for the particles must be insoluble in the developing solution; otherwise, all of the material layer may dissolve, leaving only the resist layer. The success of this development indicates that the process shown in Figure 1a,b is possible. 

Figure 6 shows the photographs, SEM images, and energy dispersive spectrometry (EDS) color-mapping images of the sintered AgI patterns using a 1:1 slurry ratio. In Figure 6a, fine light-yellow AgI patterns are clearly visible on the substrate after sintering, without a large pattern loss during the sintering process. This implies that AgI can be patterned without chemical deposition. Figure 6b presents a prospective SEM image of the AgI pattern. The AgI dots and line patterns are created as observed in the SEM images. Figure 6c shows a top view of the border area of the dots and the line pattern. Figure 6b,c show a finely photolithographed pattern successfully produced at the microscale. However, some roughness is visible on the line patterns. The top view of the dot pattern with a pitch of 400 µm, where the dots are evenly spaced, is shown in Figure 6d. The overall appearance of the pattern resembles that of a chemically deposited pattern [23]. The single-dot image in Figure 6e shows a well-fabricated circular structure. The sintered crystal density differs between the center and the edge of the dot. The high-magnification view enclosed within the yellow square line in Figure 6e is shown in Figure 6f. As shown in Figure 6f, although the particles were connected at the edge of the dot, the density was low, and the pores were visible. At the center, the particles became denser, and the grain boundaries were observed. The size of the grains increased at the center of the dot.

The line pattern with the 400-µm pitch is displayed in Figure 6g. The final shape is acceptable, although some line breaks are observed. Figure 6h shows an enlarged SEM image of a single line, where the density difference between the line edge and center is evident and similar to the dot pattern. Thus, the density difference does not depend on the shape of the pattern. Figure 6i shows that the edge of the line exhibits roughness with a low material density, whereas the density increases at the center. Furthermore, the grain boundaries were visible at the center.

There are some issues with the decomposition of AgI into Ag and I under intense light exposure. Figure 6i–l show the EDS color mapping of AgI, where Ag, I, and Si are represented in red, yellow, and blue, respectively. Figure 6j confirms that the base Si wafer is perfectly exposed between the dots and the line. The uniform mixture of red and yellow colors in Figure 6k indicates that AgI was evenly distributed in the pattern without significant decomposition. The presence of a binder coating on the AgI particles protects the AgI nanoparticles from UV radiation. The enlarged image inside the black square line in Figure 6k is shown in Figure 6l, which clearly illustrates the interface between AgI and the Si wafer at the edges of the patterns. Thus, the EDS analysis supports the successful lithographic formation of AgI patterns on the substrate.

### 3.3. Changes in the Dimensions of AgI Patterns Caused by Sintering

Particle materials undergo shrinkage and crystal growth during sintering, along with the removal of the binder and resist, making it crucial to investigate the changes in the pattern shape. Figure 7 illustrates the changes in the shape of the dot and line through sintering. The optical microscopy images in Figure 7a,d indicate a clear decrease in the size of the single-dot pattern after sintering. This size reduction is evident in the SEM images shown in Figure 7b,e, where the diameter shrinks from 69 µm to 42 µm in both the vertical and horizontal directions. As seen in Figure 7b, the dot, before sintering, is an aggregate of circular powders. However, after sintering, they transform into a single agglomerated dot, as depicted in Figure 7e, indicating that AgI particles adhere to each other during sintering. In Figure 7e, some powder particles are visible as independent particles surrounding the sintered dot, positioned inside the original dot circle before sintering. The change in the height of the dot pattern is shown in Figure 7c,f, reducing from 20 µm to less than 10 µm after sintering.

Figure 7g–l present optical-microscope and SEM images of the line pattern before and after sintering. Before sintering, the distribution of AgI powder on the pattern can be seen. The appearance of the powder on the pattern is no different from that of the dots. The width of the line before sintering was approximately 70 µm, as shown in Figure 7h, which was larger than the mask design, but after sintering, as shown in Figure 7k, it shrinks to generally 40–50 µm, depending on the location, and is about 40 µm in this part of the SEM image. Figure 7i,l show line patterns viewed from the side before and after sintering, respectively. The height of the line before sintering was lower than that of the dots, about 15 µm. After sintering, the height of the line pattern also shrinks, but as seen in Figure 7l, the surface appears rougher than that of the dots in Figure 7f.

The uniform reduction that occurred in the dot pattern suggests that the line shrank in all directions, leading to the anticipated break in the line, as can be seen in Figure 6b,f. The break in the line pattern is thought to be due to the difference in thermal expansion coefficients of AgI and the Si substrate during sintering and is not considered a problem with lithography. It is expected that the fracture of this pattern would be less likely with a lower sintering temperature. Therefore, it is recommended to attempt the optimization of the sintering temperature within the range of above 350 °C, which is necessary for the removal of SU8, to a maximum of approximately 480 °C.

Figure 8 shows a bar graph of the changes in the dot diameter, line width, and pattern pitch. The blue and orange bars indicate the measurements before and after sintering, respectively. The black dotted line in Figure 8a represents the designed diameter and width of the mask, both set at 50 µm. Figure 8a shows that the average diameter decreased by approximately 40%, suggesting that the mask should be designed with a diameter approximately 40% larger than the final diameter for precise control of the dot diameter. For the line pattern, the line width decreased by approximately 44% after sintering. The shrinkage ratio varies depending on the shape of the pattern, and it is hypothesized that the shrinkage ratio is influenced by fabrication parameters such as the AgI slurry concentration and resist polymer type. 

The bar graphs in Figure 8b show changes in the line and dot pitches, with the blue and orange bars representing the values before and after sintering. The values in the parentheses on the vertical axis indicate the design pitch of the mask (Figure 2). Neither the dot nor the line pitches of the sintered patterns deviate from the designed mask values. While the pitch value includes errors in the SEM’s working distance, it is acceptable to consider that the pitch has not changed. From the above, it can be said that in order to make the diameter or width the desired dimension, it is necessary to design the mask 40–44% larger, while the pitch can be designed according to the desired dimension.

Enlarged SEM images of the pattern edge and center, both before and after sintering, are presented in Figure 9. The left and right sides of Figure 9 display the dot or line edge and center, respectively. Figure 9a,b, captured before sintering, show that the AgI particles are attached to each other and cover the edge of the dot SU8 resist. After sintering, as depicted in Figure 9c,d, the AgI particles are fully connected, and the size of the AgI crystal has increased. The appearance of the sintered AgI is similar to that of other sintered materials such as ceramics [36,37]. Moreover, the dot edge and center areas exhibit different densities. At the edge, the low-density particle-like crystals are interconnected, and some pores can be observed. Conversely, in the center area, the crystal grows larger than at the edge, and a grain boundary can be discerned.

The cases of the edge and center of the line pattern were similar to those of the dot pattern. As shown in Figure 9e,f which are the edge and the center of line before sintering, there is almost no difference in the appearance compared to the dot, but the line edge seems to have fewer particles. It is speculated that when the line was removed from the developer, the developer flowed along the line, removing the particles. Figure 9g,h depict the edge and center of the line after sintering, respectively. Here too, low-density crystals can be seen at the edges, and the center has transformed into a large crystal with more clearly visible grain boundaries than the dots. In any case, the AgI particles grew into crystals through sintering, and were able to form a pattern of AgI crystals. However, both dots and lines exhibited differences in crystal shape between the edge and the center, suggesting that this is not dependent on the pattern shape, but is influenced by the process. Therefore, we considered the pattern formation mechanism from UV exposure to development and sintering.

The sintering mechanism is illustrated schematically in Figure 10. As shown in Figure 10a, the SU8 layer is cured inside the white dotted lines by UV diffusion. The UV spreading exceeds the cured area at the top of the SU8 layer, as evidenced by the diameter of the dot pattern before sintering being larger than the mask design with a diameter of 50 µm. After development, the AgI layer protrudes from SU8, as shown in Figure 10b. This can be seen in Figure 7c,i, in which AgI particles cover the side of the SU8 layer. During sintering, the binder and SU8 were removed, allowing AgI to come into direct contact with the Si wafer. After sintering, the AgI particles in the protruding region formed a low-density structure at the outer edge of the pattern as shown in Figure 10c.

### 3.4. Mechanical Adhesion Strength of AgI Pattern

The adhesion strength of the sintered AgI pattern to the Si wafer was confirmed through a tape test. The tape, fully adhered to the AgI pattern, was peeled off, and the presence or absence of pattern peeling was checked. Figure 11 shows photographs of the peeling process and microscope images. As depicted in Figure 11a,b, no significant peeling of the pattern occurred when the tape adhered to the AgI pattern was peeled off. The pattern appears to be slightly thinner as shown in Figure 11b, which is thought to be due to the removal of the AgI stacked on the AgI and the removal of the AgI contaminant particles on the Si wafer. However, the AgI pattern itself was not removed from the substrate. From the optical microscope image in Figure 11c, no difference can be observed between the part where the tape was peeled off (left side of the red dotted line) and the original part (right side of the red dotted line). The contaminant particles between the patterns are reduced in the area to the left of the red dotted line. Therefore, it can be inferred that the sintered AgI adheres to the Si substrate with a force stronger than the adhesive force of this tape. The strength of the pattern in the frost generation experiment is sufficient, and there is a minimal possibility of it being damaged by ice or washed away by water.

### 3.5. In-Situ Demonstration of Frosting on the AgI Patterns

The in-situ observation of condensation frosting on the fabricated AgI dot was demonstrated using the experimental setup depicted in Figure 12a. Figure 12b–g show images captured from a video illustrating the formation of condensed droplets and frost growth from a single AgI dot. In Figure 12b, frost emerged from the surface of the AgI dot, as indicated by the two yellow arrows. On the Si substrate surface, numerous condensed water droplets with diameters of approximately 10–20 µm are observed. As shown in Figure 12c, 8 s after Figure 12b, the ice arms extend towards the nearby droplets. The thick arrow indicates the moment when the ice arm comes into contact with a neighboring droplet and integrates into the frost tip. Figure 12d captures the ice arm 16 s after Figure 12b, clearly showing this process with thin arrows indicating ice bridges between the droplets. The thick arrows signify the conversion of the surrounding water droplets into ice. The freezing of droplets upon contact with the ice arm suggests that the water droplets were in a supercooled state, which is consistent with previous observations of ice-bridging phenomena [13,14,15].

Figure 12e–g illustrate different behaviors from those presented in Figure 12b–d, indicating the absorption of humidity by the growing frost. As shown in Figure 12e, a frost arm is formed on the surface of the AgI dots from the point indicated by the yellow arrow. Numerous condensed water droplets were observed within the region enclosed by the blue dotted line. Figure 12f shows that droplets in the blue box in Figure 12e decreased in size or disappeared after 8 s. Figure 12g, captured 16 s after Figure 12e, reveals that most surrounding droplets evaporated and contributed to frost growth. Eventually, the surrounding droplets vanished completely, and the growth of the frost arm slowed down noticeably. In the behavior depicted in Figure 12e–g, the ice bridge does not form until the ice arm comes into contact with the droplets. This phenomenon suggests that the frost arm absorbs the surrounding humidity from the droplets, thereby agreeing with previous studies [14,24].

In-situ observation photos of frost formation on the line pattern are presented in Figure 13. Figure 13a,b depict the state of frosting on a 200 µm pitch, while Figure 13c,d illustrate the frosting on an 800 µm pitch. The images were captured 1 and 10 min after the start of sample cooling, respectively. In Figure 13a,b, condensation droplets appear between the AgI lines in both cases, but the sizes of the droplets differ. A comparison of Figure 13b,d after 10 min reveals differences in the shape of the ice formed on the AgI. Hexagonal or quadrangular frost is formed in Figure 13b, while needle-shaped frost is formed in Figure 13d. Additionally, in Figure 13b, no ice grains are seen between the AgI lines, whereas in Figure 13d, ice grains that are not touching the AgI lines are formed. This experiment was conducted in a constant temperature and high-humidity chamber, and similar results were reproduced twice. As these results show, the shape and behavior of frost formation vary depending on the arrangement of the AgI lines. There is no previous research observing frost formation by intentionally changing the regular arrangement of AgI, such as this line and space, which demonstrates the advantage of the flexibility of this patterning method. In summary, the results demonstrate that AgI patterns, fabricated using the current method, can induce frost growth. Basic frosting phenomena, such as ice bridges and ice arms, are consistent with results from previous frost studies. Frosting observed in the UV-exposed AgI pattern suggests that UV light energy does not alter the properties of AgI as a starting point for frost formation. The shape of the AgI crystal structure could affect frost growth, necessitating further studies on frost structures in the future. This method is valuable for studying frost growth as it enables flexible patterning of frost generation points using AgI across a wide area.

## 4. Conclusions

We introduced a photolithography-based method for patterning particle materials on a substrate, with specific emphasis on ionic crystal materials such as AgI, which is known to induce frost formation. AgI was patterned on a Si substrate to provide valuable insights into the mechanism of frost formation. The proposed method involved stacking layers of the AgI slurry and SU8, simultaneously exposing them to UV light and developing. The size of patterns before sintering expanded more than the mask design value due to UV spreading. However, AgI pattern changed in size during sintering. For the dot pattern, the diameter before sintering exceeded the mask design diameter of 50 µm. After sintering, the diameter decreased by approximately 40%, resulting in a particle size smaller than the design value. The line width also underwent an approximately 44% shrinkage. Therefore, the dimensions of the parts to be sintered later, such as the diameter and width of AgI, need to be designed taking into account the shrinkage. Despite variations in pattern size before and after sintering, pitches of both patterns remained unchanged from the mask design values indicating that the pattern pitch can be fabricated precisely to match the design values. Finally, condensation frosting was observed in-situ using the fabricated AgI dot. The formation of ice bridges and the absorption of surrounding water droplets are consistent with previous research, demonstrating the effectiveness of the proposed patterning method for ice-leading material. Therefore, this method allows for the fabrication of AgI patterns without affecting the crystal structure, overcoming the challenges of large-area fabrication and the limitations of free pattern fabrication that are barriers in conventional approaches. As a future prospect of this research, it could be useful for studying the differences in frost behavior between AgI patterns and research on ice prevention due to pattern differences. We believe that this method, with its high degree of pattern freedom, large area, and high throughput due to the application of photolithography, will be useful in the study of frost prevention and defrosting. Furthermore, the proposed method allows for large-area and high-throughput fabrication of particle materials, providing a new approach for manufacturing functional surfaces. For example, patterning of ceramics can be expected to improve the performance of solid oxide fuel cells. With regard to the resolution of this method, it has been possible to fabricate without any problems if the pitch is sufficiently wide for a diameter or line width of 50 µm, as in the case of the mask used in this experiment. However, currently, for patterns smaller than this, for example, 20 µm dots, with a mask with a pitch of 40 µm, the AgI particles tend to stick to each other and are not properly developed. The resolution of this method is dependent on the particle size, making it a future task to determine the minimum pattern size and width using nanoparticles smaller than those used in this experiment, and to investigate the resolution.

## Figures and Tables

**Figure 1 nanomaterials-13-03035-f001:**
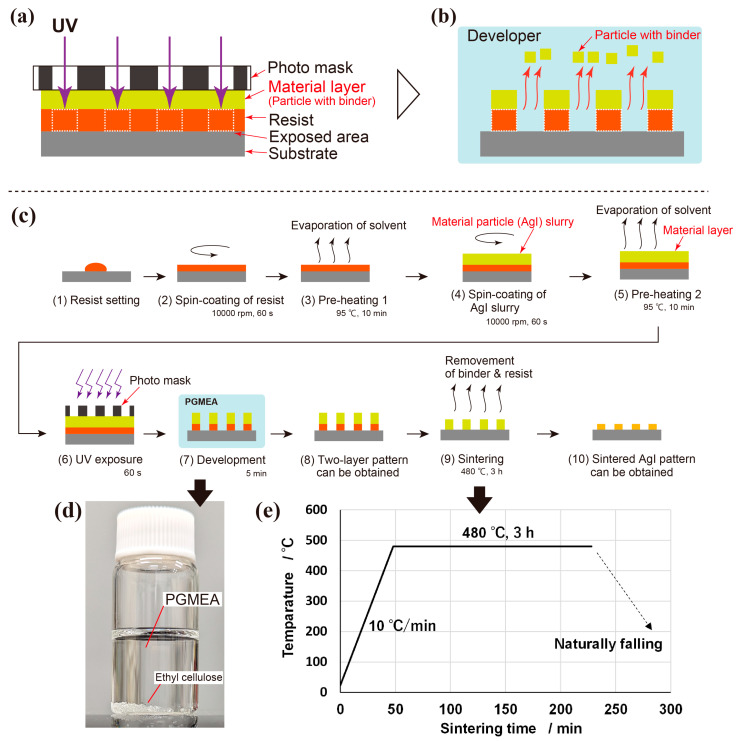
Proposed material patterning process using UV photolithography: (**a**) UV exposure of photoresist through a compound material layer of particles and organic binder, (**b**) development of material layer with photoresist, (**c**) details of entire fabrication process, (**d**) ethyl cellulose can stand in the PGMEA, (**e**) sintering temperature program.

**Figure 2 nanomaterials-13-03035-f002:**
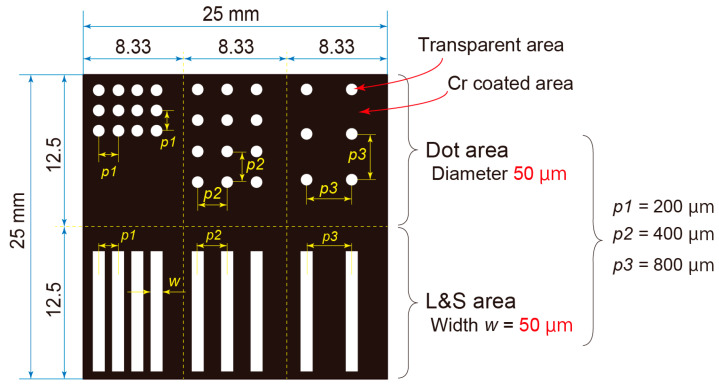
Design of the photomask with six patterned areas.

**Figure 3 nanomaterials-13-03035-f003:**
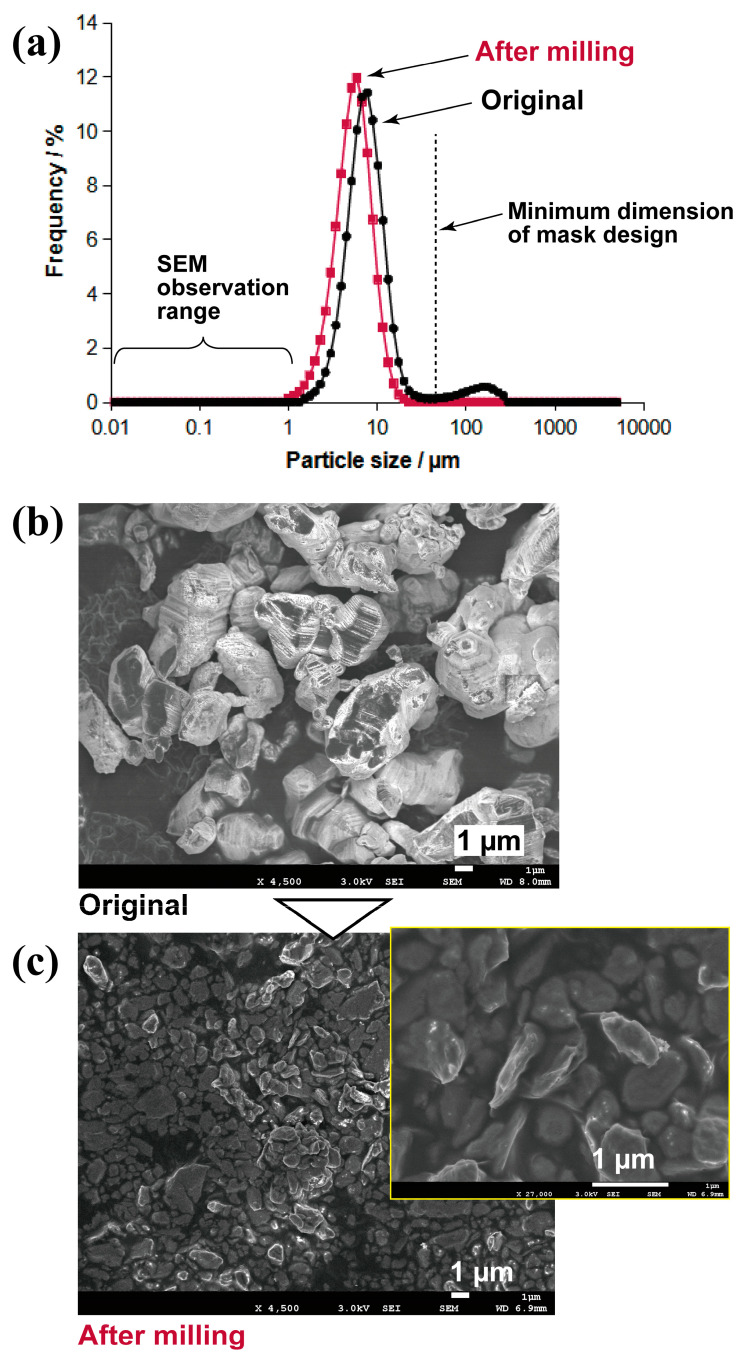
(**a**) Size distributions of AgI particles before and after ball milling; SEM images of particles (**b**) before and (**c**) after milling.

**Figure 4 nanomaterials-13-03035-f004:**
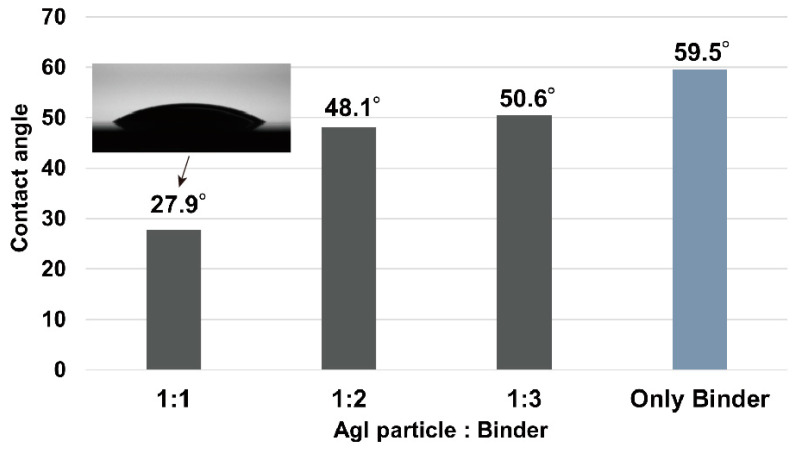
Contact angles of the slurries on the pre-exposed resist (SU8) layer.

**Figure 5 nanomaterials-13-03035-f005:**
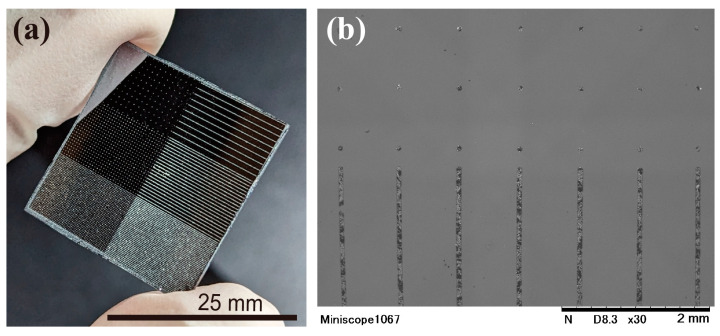
Before-sintering sample. (**a**) Photo of the sample that has light-yellow AgI dots and line patterns on the Si substrate; (**b**) SEM image of the dot and line pattern.

**Figure 6 nanomaterials-13-03035-f006:**
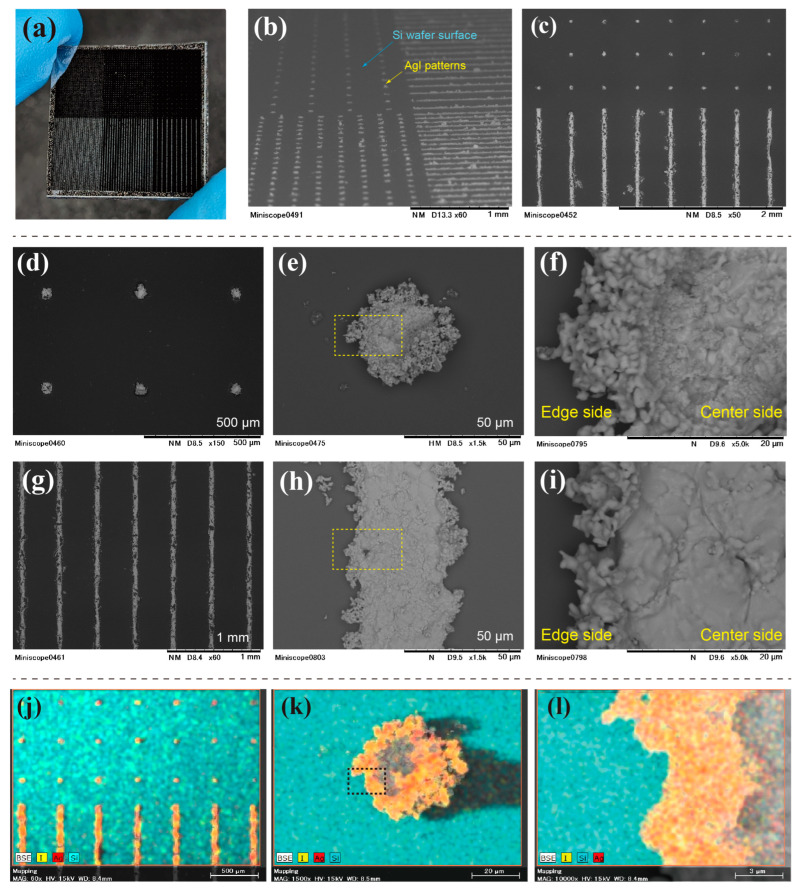
Photo, SEM, and EDS color-mapping images from after AgI patterns had been sintered; (**a**) photo of sintered AgI pattern sample, (**b**) perspective image of the AgI-patterned Si substrate, (**c**) top-view of the dot and line areas, (**d**) AgI dot pattern with the designed pitch of 400 μm, (**e**) single dot image, (**f**) enlarged image of the dot, (**g**) AgI line pattern with 400 μm pitch, (**h**) single line image, (**i**) enlarged image of the line, (**j**) EDS color-mapping images of the Ag I pattern, (**k**) EDS image of the single AgI dot, and (**l**) its enlarged image.

**Figure 7 nanomaterials-13-03035-f007:**
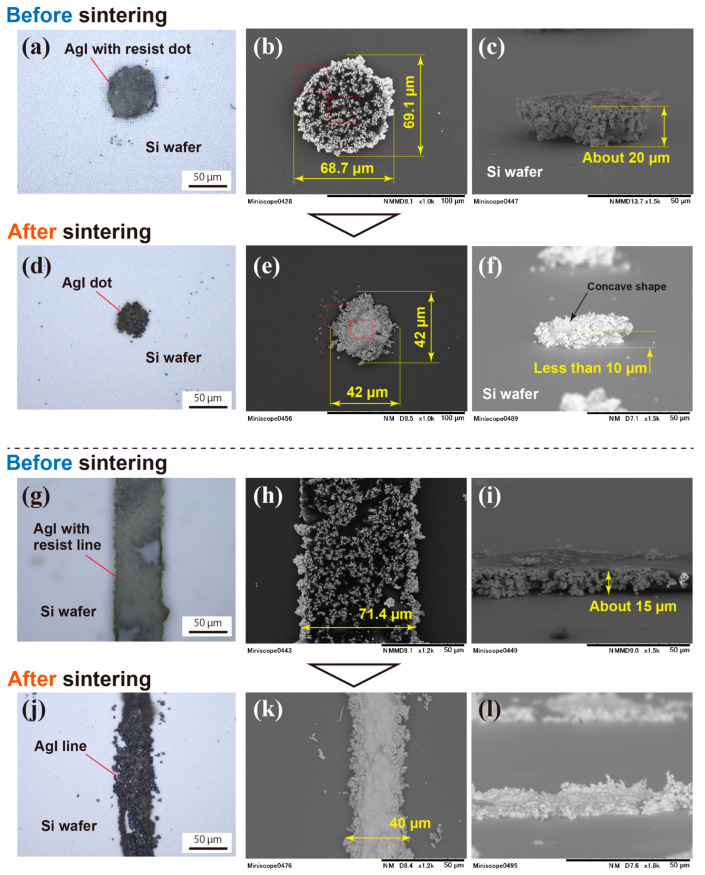
Shapes of dot and line before and after sintering; (**a**) optical microscope image of the dot before sintering, (**b**) SEM top-view image of the single dot before sintering, (**c**) SEM side view of the dot before sintering, (**d**) optical microscope top-view image of sintered dot, (**e**) SEM image of the sintered dot, (**f**) SEM side view of sintered dot, (**g**–**i**) optical microscope, top-view SEM and side-view SEM image of the line before sintering, and (**j**–**l**) optical microscope, top-view SEM and side-view SEM image of the AgI line after sintering.

**Figure 8 nanomaterials-13-03035-f008:**
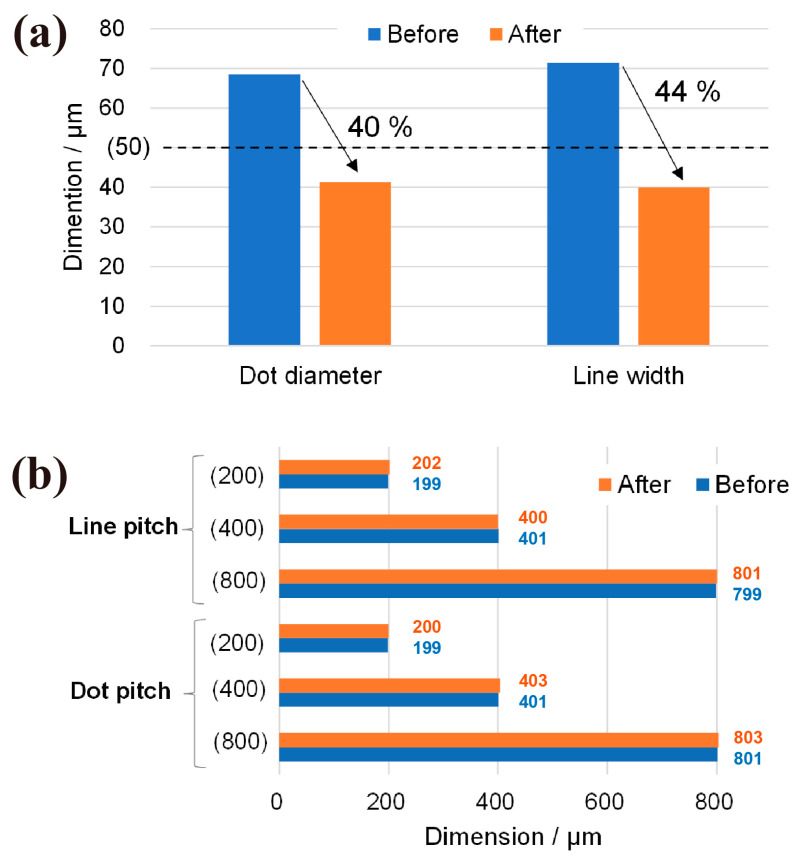
Dimension changes through sintering process measured in Figure 7; (**a**) changes in dot diameter and line width, and (**b**) pitches.

**Figure 9 nanomaterials-13-03035-f009:**
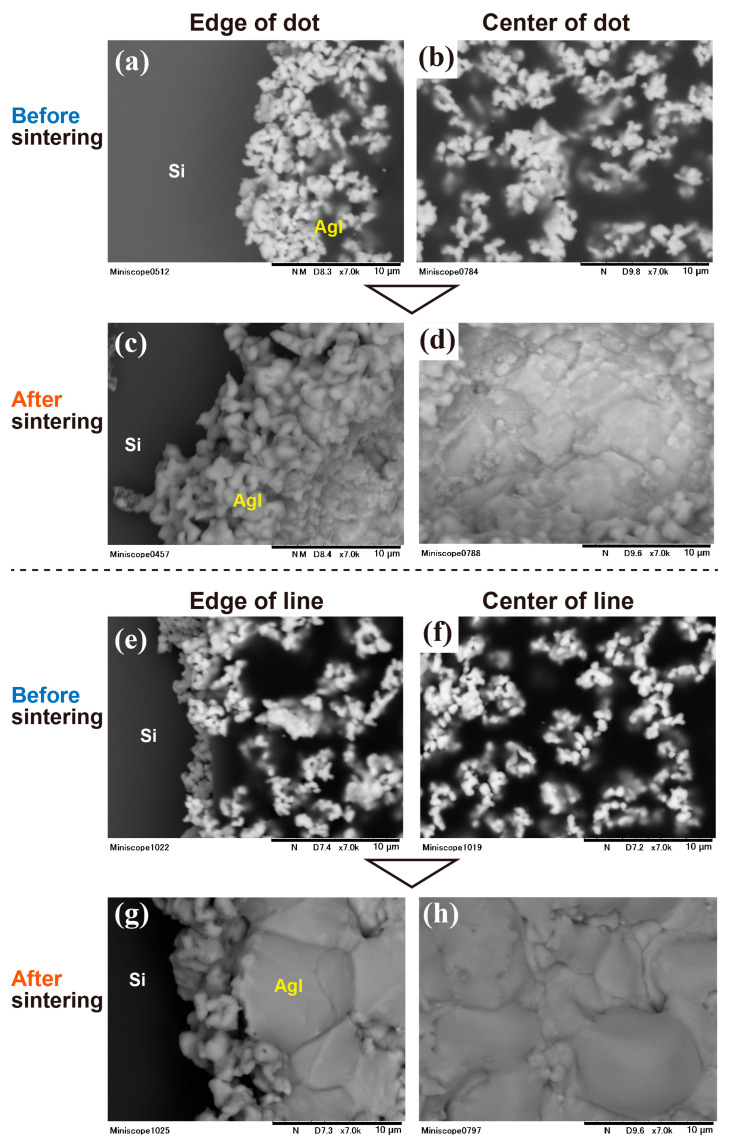
SEM image of edge and center of the AgI dot and line before and after sintering; (**a**,**b**) dot edge and center before sintering, (**c**,**d**) sintered dot edge and center, (**e**,**f**) line edge and center before sintering, and (**g**,**h**) sintered line edge and center.

**Figure 10 nanomaterials-13-03035-f010:**
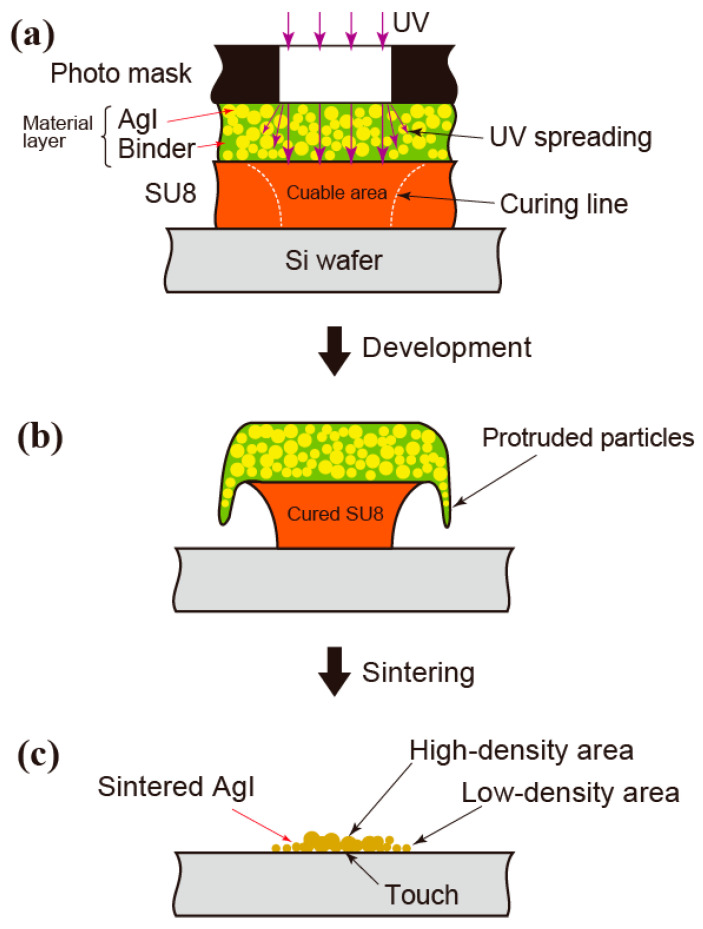
Schematic of AgI pattern formation process: (**a**) UV curing of resist layer through the material layer, (**b**) after development of resist with protruded material layer, and (**c**) AgI after sintering.

**Figure 11 nanomaterials-13-03035-f011:**
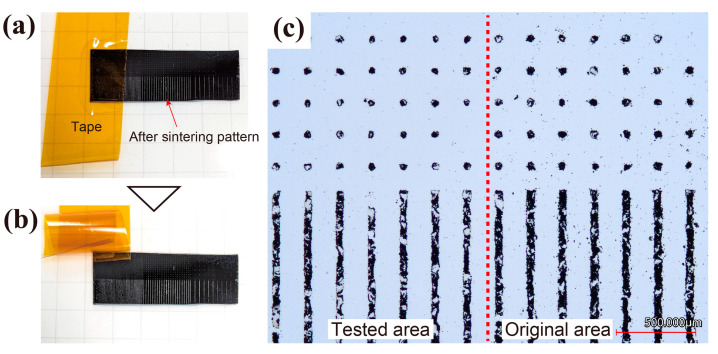
Mechanical adhesion testing of sintered AgI pattern: (**a**) taping on the AgI pattern, (**b**) removing of the tape, and (**c**) optical microscope photo of the tested and non-taped area.

**Figure 12 nanomaterials-13-03035-f012:**
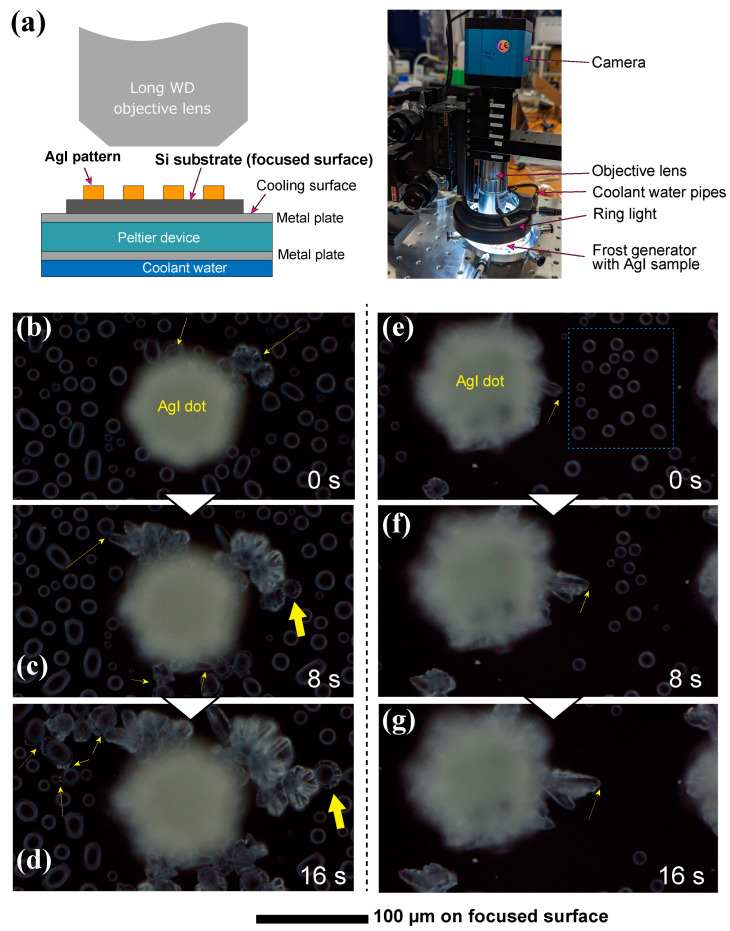
In-situ observation setup and frost-growth phenomenon from the fabricated AgI dot pattern: (**a**) schematic of the experimental setup, (**b**) frost spreading behavior through the ice-bridge, (**c**) ice-pillar growth absorbing vapor from surrounding condensed droplets, (**d**) connection of droplets by ice bridge and changing of droplet to ice, (**e**) growing of ice arm to the condensed droplets, (**f**) shrinkage of droplets with growing the ice arm, and (**g**) disappearance of droplets.

**Figure 13 nanomaterials-13-03035-f013:**
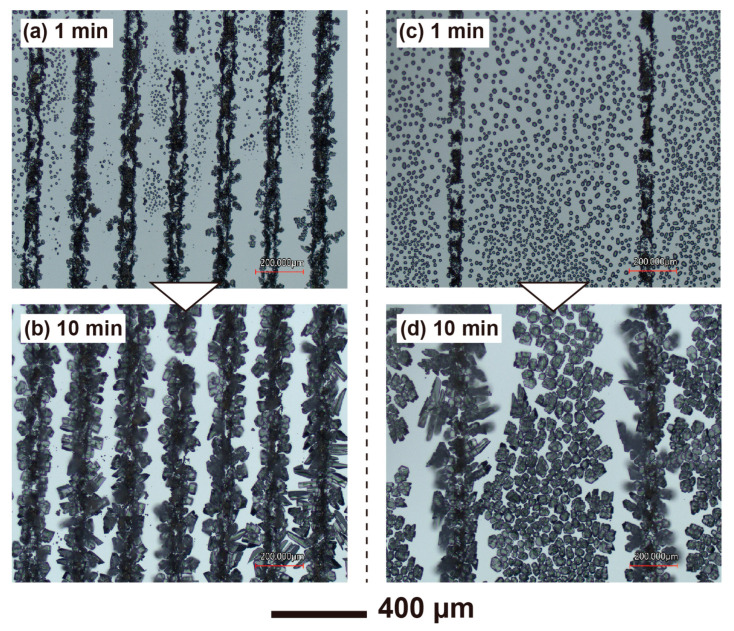
Frost growth phenomenon on the fabricated AgI line pattern: (**a**,**b**) on the 200 µm pitch at 1 and 10 min after start cooling, and (**c**,**d**) on the 800 µm pitch at 1 and 10 min after start cooling.

## Data Availability

Data are contained within the article.

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
