# Peer review of "Fabrication of Silver Iodide (AgI) Patterns via Photolithography and Its Application to In-Situ Observation of Condensation Frosting"

_nanomaterials, 2023, doi:10.3390/nano13233035_

Round 1

Reviewer 1 Report

Comments and Suggestions for Authors

Subject: Review of Manuscript Submission Nanomaterials_ 2723383

I have carefully reviewed the manuscript titled "Fabrication of Silver Iodide (AgI) Patterns via Photolithography and Its Application to the In-Situ Observation of Condensation Frosting" submitted to Nanomaterials for consideration.

This study introduces an innovative photolithography-based method for patterning ionic and inorganic particle materials such as silver iodide (AgI). This versatile method applies to large areas with high throughput and presents new opportunities for modifying functional surfaces. I think this article is meaningful. It can be accepted after revision.

1.     The manuscript presents a novel photolithography-based method for patterning ionic materials, specifically AgI. The authors are commended for addressing the challenges in precision associated with conventional patterning of powdered materials. However, it would be beneficial if the authors could provide a comparative analysis demonstrating the precision of the proposed method relative to existing methods. This would help quantify the improvement in patterning resolution and detail the practical limits of the technique.

2.     While the method for creating well-defined AgI patterns is described, further characterization of these structures would strengthen the manuscript. The inclusion of SEM or TEM images post-sintering could provide visual confirmation of the pattern fidelity and the binder removal efficiency. Additionally, an assessment of the mechanical integrity of the AgI patterns on the Si substrate could be valuable for understanding their durability in practical applications.

3.     The sintering step appears to be critical for the removal of binders and ensuring the adhesion of the AgI patterns. The manuscript would benefit from a more detailed description of the sintering conditions, such as temperature profiles, atmosphere, and time. Moreover, it would be informative to include how these parameters were optimized and how they influence the final pattern quality.

4.     The in-situ observation of condensation frosting on the AgI patterns is a key application of the proposed method. The authors should elaborate on the experimental setup for these observations, including control of environmental conditions, imaging techniques, and reproducibility of the frosting patterns. Clarification on how this method compares to the limitations of previous techniques in frost research would further validate the study's contributions.

Reviewer 2 Report

Comments and Suggestions for Authors

The paper “Fabrication of Silver Iodide (AgI) Patterns via Photolithography and Its Application to the In-Situ Observation of Condensation Frosting” deals with a new method for directly processing ionic materials on substrates.

It would be advantageous if the materials and methods were presented initially. For example, the negative tone resist is mentioned earlier in the process description but not described until later. The devices and test methods used are only mentioned later in the results. In this context, it also needs to be clarified how measurements of the contact angle were carried out; three different concentrations are mentioned in the text - does this mean the ratio of AgI to the binder - but not which measurement volume was used?

The statements based on Figures 5b) and c) are poorly substantiated; the figures only show the qualitative behavior of the solution. The shrinkage of the particles during sintering is shown in Fig. 7. Instead of the diameter, a measurement of the area (and possibly an equivalent diameter) would be more appropriate, as the irregular shape would have less influence on the diameter. The experiments on condensation frosting are interesting, but information on the temperatures, ambient conditions, etc., would also be desirable.

Reviewer 3 Report

Comments and Suggestions for Authors

Authors report silver iodide (AgI) patterns using photolithography. It is difficult to make inorganic materials fine patterns. Vacuum evaporation and shadow mask can be used to make inorganic materials fine patterns but the shape and size will be limited. Authors suggested interesting method for ionic material’s patterns.

I recommend this manuscript to Nanomaterials after a minor revision.

1.     Line 318-323, Figure 9(b) : Is there some difference between before and after dimension? In the Figure 9(b), the blue bars seem slightly smaller than orange bars,

2.     Method, fabrication and characterization are very good. Application and demonstration are not so much attractive to me. I know this platform can be useful the research of “Cloud seeding”. Can authors explain the value of their work in the introduction and discussion.

Round 2

Reviewer 1 Report

Comments and Suggestions for Authors

I am satisfied with the revision. It can be accepted in its present form.